# Controlled Oxidation of Cobalt Nanoparticles to Obtain Co/CoO/Co_3_O_4_ Composites with Different Co Content

**DOI:** 10.3390/nano12152523

**Published:** 2022-07-22

**Authors:** Aleksandr S. Lozhkomoev, Alexander V. Pervikov, Sergey O. Kazantsev, Konstantin V. Suliz, Roman V. Veselovskiy, Andrey A. Miller, Marat I. Lerner

**Affiliations:** 1Laboratory of Nanobioengineering, Institute of Strength Physics and Materials Science, Siberian Branch of the Russian Academy of Sciences, 634021 Tomsk, Russia; kzso@ispms.tsc.ru (S.O.K.); konstantin.suliz@gmail.com (K.V.S.); 2Laboratory of Physical Chemistry of Ultrafine Materials, Institute of Strength Physics and Materials Science, Siberian Branch of the Russian Academy of Sciences, 634021 Tomsk, Russia; pervikov@list.ru (A.V.P.); lerner@ispms.tsc.ru (M.I.L.); 3Research and Education Center of Additive Technologies, National Research Tomsk State University, 634050 Tomsk, Russia; 4Schmidt Institute of Physics of the Earth, Russian Academy of Sciences, 123242 Moscow, Russia; roman.veselovskiy@ya.ru; 5Shared Use Center “Nanotech”, Institute of Strength Physics and Materials Science, Siberian Branch of the Russian Academy of Sciences, 634021 Tomsk, Russia; miller@ispms.ru

**Keywords:** electrical explosion of wire, cobalt nanoparticles, oxidation, composite nanoparticles, magnetic properties

## Abstract

The paper studies patterns of interaction of electroexplosive Co nanoparticles with air oxygen during heating. The characteristics of Co nanoparticles and composite Co/CoO/Co_3_O_4_ nanoparticles formed as a result of oxidation were studied using transmission electron microscopy, X-ray phase analysis, thermogravimetric analysis, differential scanning calorimetry, and vibrating sample magnetometry. It was established that nanoparticles with similar morphology in the form of hollow spheres with different content of Co, CoO, and Co_3_O_4_ can be produced by varying oxidation temperatures. The influence of the composition of composite nanoparticles on their magnetic characteristics is shown.

## 1. Introduction

Cobalt is a transition metal that has a beneficial effect on human health [1,2]. It is contained in vitamin B12 which is beneficial in the treatment of anaemia since it provokes the formation of red blood cells [2]. Cobalt has unique magnetic, optical, electrical, and catalytic characteristics which make it suitable for a wide range of applications in the field of nanoelectronics and nanosensors [3,4,5].

Co_3_O_4_ is a multifunctional material and has many applications, such as biomedical applications [6], gas sensors [7], solar selective absorbers [8], anode materials in lithium-ion batteries [9], energy storage [10], field emission materials [11], heterogeneous catalysis [12] and others.

CoO is successfully used in catalytic processes of CO_2_ hydrogenation [12], as photo- and electrocatalytic materials [13,14].

The unique properties of cobalt and its oxides are also used in Co/CoO, Co/Co_3_O_4_ and CoO/Co_3_O_4_ systems [15,16,17,18,19,20]. Such composite nanoparticles are obtained by chemical methods based on the oxidation of organometallic precursors [21,22], using solvothermal methods [23], oxidation of Co nanoparticles with water [24], and green chemistry methods [25,26].

Controlled oxidation of Co nanoparticles with oxygen during heating can become a promising method for the production of composite nanoparticles based on cobalt and its oxides. As shown in [27], cobalt nanoparticles heated in the air form an amorphous CoO layer on the surface, which grows over time through an indirect mechanism of exchange followed by oxidation to Co_3_O_4_.

Controlled oxidation of cobalt nanoparticles can contribute to the production of composites based on cobalt and cobalt oxides with a given component ratio. This is important both in terms of the evaluation of the magnetic characteristics, for example, to ensure the exchange anisotropy of the nanoparticles at the ferromagnetic/antiferromagnetic interface [28], and in terms of catalytic activity [24,29,30].

As a precursor for obtaining such composites, cobalt nanopowder obtained by electrical explosion of cobalt wire in an inert medium is promising [31]. The method is quite productive, one setup provides productivity up to 200 g/hour [32]. The purity of the product is determined by the composition of the initial wire and buffer gas of the inert medium. 

The preparation of the cobalt compound and cobalt-oxide-based composites using Co nanopowders has not been considered. In this regard, the purpose of this paper is to study the patterns of oxidation with air oxygen by heating Co nanoparticles obtained by electrical explosion of wire and to determine the effect of temperature on the composition, morphology and magnetic properties of oxidation products.

## 2. Materials and Methods

Co nanoparticles were obtained using a setup described in [33] at electrical explosions of cobalt wire (chemical purity of 99.98%) with a diameter of 0.5 mm in argon (chemical purity of 99.993%), at a voltage of 29 kV and a capacity of 3.2 μF. The electrical schematic of the setup and the current and voltage recording are given in [34].

The morphology of nanoparticles was studied by transmission electron microscopy (TEM) using a JEM-2100 microscope (JEOL, Tokyo, Japan). The average size of nanoparticles was determined by particle size distribution histograms obtained from electron microscopy data. To construct a histogram, 2842 particle diameters were measured. The average size was determined by the expression an = Σniai/Σni, where ni is the number of particles that fell into the selected size range, ai is the average diameter of the particles in the selected interval.

Oxidation of Co nanoparticles was studied by methods of thermogravimetric analysis and differential scanning calorimetry (TG-DSC) using NETZSCH STA 449F3 (Netzsch, Waldkraiburg, Germany). For this, 5 mg samples were heated in airflow from ambient temperature to 760 °C at a heating rate of 10 °C/min.

Composite nanoparticles were obtained by heating the Co nanopowder in a muffle furnace up to a range of temperatures (150, 250, 300, 450 or 600 °C) at a heating rate of 10 °C/min, kept for 2 h and cooled to room temperature.

The phase composition of nanoparticles was determined using a Shimadzu XRD 6000 X-ray diffractometer (Shimadzu, Kyoto, Japan). The obtained data were processed with Powder Cell 2.4 software (W. Kraus& G.Nolze, Berlin, Germany).

Magnetic properties of nanoparticles were studied at the Centre for Collective Use of the IPE RAS [35] using a PMC MicroMag 3900 vibrating sample magnetometer (Lake Shore Cryotronics, Westerville, OH, USA) at room temperature in air. Parameters of the hysteresis loop, such as coercive force (Hc), residual saturation magnetisation (Mr), and saturation magnetisation (Ms), were measured when the sample was magnetised in a magnetic field up to 15,000 E.

## 3. Results and Discussion

Co nanoparticles obtained by electrical explosion of wire are spherical nanoparticles with an average size of 56 nm, which are covered with a solid oxide film with a thickness of about 2.5 nm, formed as a result of passivation of nanoparticles in air (Figure 1a–c). The peaks on the diffraction pattern (Figure 1d) correspond to planes (111), (200) and (220) in accordance with a powder diffraction file of the International Centre for Diffraction Data (ICDD), card No. 00-015-0806.

It is known that Co is well oxidised by oxygen when heated, making it possible to produce composites based on cobalt and its oxides with different contents of components. The analysis of DSC-TG curves presented in Figure 2 shows that oxidation of nanoparticles begins at the temperature of 165 °C, accompanied by an increase of the sample mass observed on the thermogravimetric curve and the beginning of an exothermic oxidation reaction. In comparison with bulk cobalt with an oxidation temperature of about 300 °C, oxidation of nanoparticles occurs at a much lower temperature. The maximum rate of the oxidation reaction is reached at a temperature of 303 °C at which a peak is recorded on the DSC curve due to the exothermic oxidation reaction. Further heating is also accompanied by an increase in the sample mass and the appearance of another endothermic peak at 462 °C, which may be associated with the oxidation of CoO to Co_3_O_4_. A noticeable decrease in the oxidation rate observed by a decreasing slope of the TGA curve occurs at 550 °C. The reduced rate asymptotically approaches the constant value towards the higher temperatures, indicating the complete oxidation of the sample. The increase of sample mass was 31.9%, which is slightly lower than a theoretical weight increase of 36.2% for completely pure metal. It may be associated with the presence of cobalt oxide in the initial sample in the form of the film on the surface of nanoparticles, which can reach ~10 vol.% at an oxide film thickness of 2.5 nm.

According to the X-ray phase analysis (Figure 3a), Co nanoparticles (ICDD PDF card No. 00-015-0806) are not oxidised at a temperature of 150 °C; there are no peaks corresponding to cobalt oxides on the diffraction pattern of the sample. Even after heating up to 250 °C, the peaks characteristic for CoO (ICDD PDF card No. 00-048-1719) and Co_3_O_4_ (ICDD PDF card No. 00-043-1003) oxides are determined in the samples in addition to the Co peaks. Further heating leads to a decrease in the intensity of the Co peaks in relation to oxides. Complete oxidation of the metal occurs at 600 °C. CoO is also oxidised to Co_3_O_4_ at this temperature. Quantitative X-ray phase analysis indicates exponential oxidation of the metal (Figure 3b, Table 1), which may be attributed to the diffusion limitations arising from the growing thickness of the oxide layer on the surface of the reacting particles.

The morphology of Co particles undergoes significant changes after heating to 250 °C (Figure 4). Voids, which are typical for metals oxidised by air oxygen during heating, are formed in the particles at the metal/oxide interface—the so-called Kirkendall effect [36,37,38]. An increase in the heating temperature leads to the gradual removal of the metal from cavities in the formed particles and the sintering of particle fragments with the formation of ~100 nm crystallites (Figure 4). The formation of hollow cobalt particles is considered in detail in [38]. The formation of such morphology is caused by thermally activated diffusion of cobalt cations through the oxide layer, followed by oxidation.

As the content of metallic cobalt in the samples decreases, their magnetic characteristics decrease (Figure 5). Cobalt nanoparticles have the highest saturation magnetisation (Ms), which was 90 emu/g. With an increase in the content of cobalt oxides in the samples, Ms decreases to 1.1 emu/g. Residual magnetisation (Mr) also decreases from 64 emu/g to 0.1 emu/g. Coercive force (Hc) decreases from 225 Oe to 51 Oe (Table 2).

The obtained samples exhibit the properties of magnetically soft materials, which is due to a narrow hysteresis loop and small coercive force at 25 °C (Figure 5a). The dependence of saturation magnetisation on the cobalt content in the samples can be described by a direct dependence (Figure 5b).

Thus, controlled heating of Co nanoparticles can be used to obtain composite Co/CoO/Co_3_O_4_ nanoparticles with similar morphology, different component contents and magnetic characteristics. According to the literature data, the magnetic characteristics of nanoparticles based on cobalt and its oxides are very different, which depends on composition, morphology, particle size and presence of impurities (Table 3). In most cases, the characteristics of nanoparticles based on Co and its oxides cannot be varied due to process parameters. In this context, the proposed method is quite attractive.

## 4. Conclusions

In the study, Co nanoparticles with a face-centred cubic crystal system and an average particle size of 56 nm were obtained by electrical explosion of wire. It was established that, when nanoparticles are heated in the air, gradual oxidation of cobalt occurs with the release of heat and the formation of CoO and Co_3_O_4_. This makes it possible to vary the ratio of metal and oxides in resulting composite Co/CoO/Co_3_O_4_ particles. Complete oxidation of Co nanoparticles occurs at 600 °C. Co/CoO/Co_3_O_4_ particles have a hollow structure attributed to the diffusion of cobalt into an oxide shell during heating. It was shown that the dependence of the saturation magnetisation of nanoparticles on the Co content is a linear dependence with a high degree of correlation. As the Co content decreases in nanoparticles, saturation magnetisation decreases from 90 emu/g at 100% Co content to 7.9 emu/g at 9.6% Co content and to 1.1 emu/g with complete metal oxidation.

The results obtained are important for technologies using such cobalt-based composites, as they allow varying the component ratio and provide multiple interfaces that determine the efficiency of such composites in photochemical, electrochemical, catalytic, and other processes.

## Figures and Tables

**Figure 1 nanomaterials-12-02523-f001:**
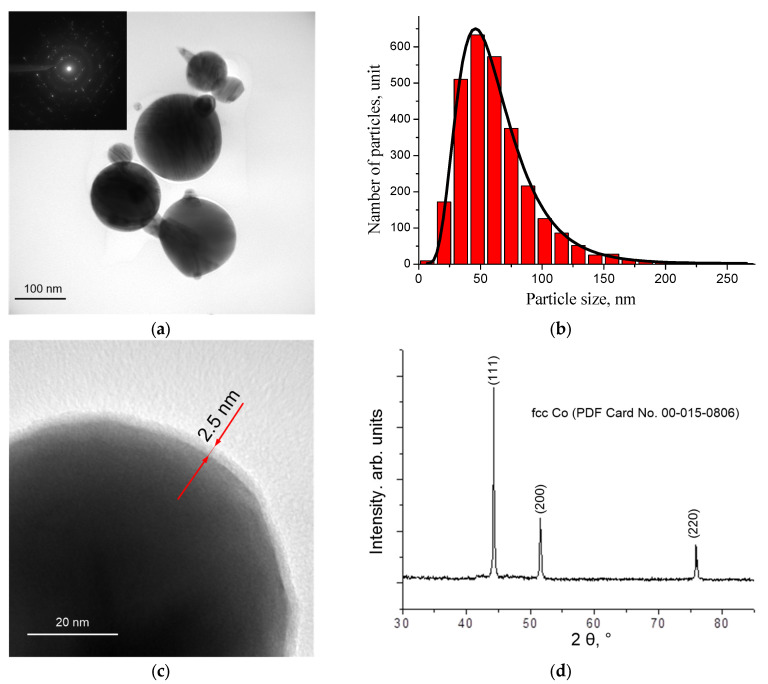
TEM image of cobalt nanoparticles (**a**), particle size distribution (**b**), TEM image of cobalt nanoparticles coated with oxide film (**c**) and X-ray phase analysis data (**d**).

**Figure 2 nanomaterials-12-02523-f002:**
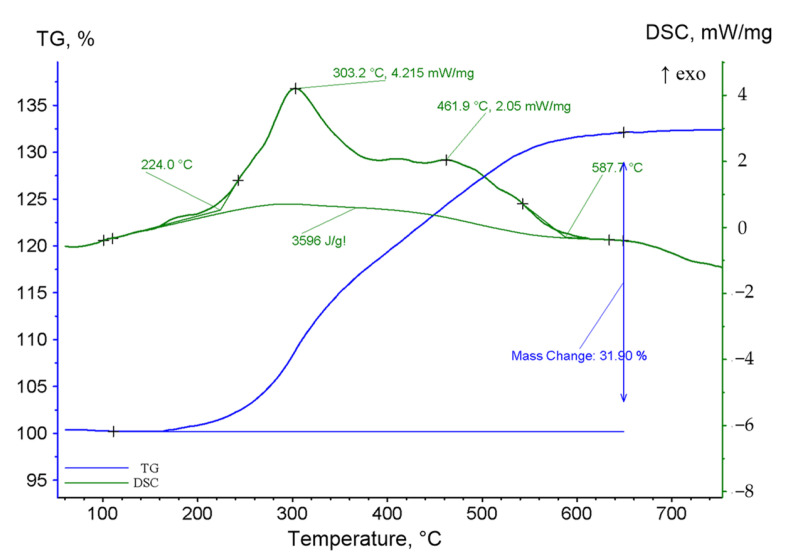
DSC-TG curves of cobalt nanoparticles when heated in an oxygen-containing atmosphere.

**Figure 3 nanomaterials-12-02523-f003:**
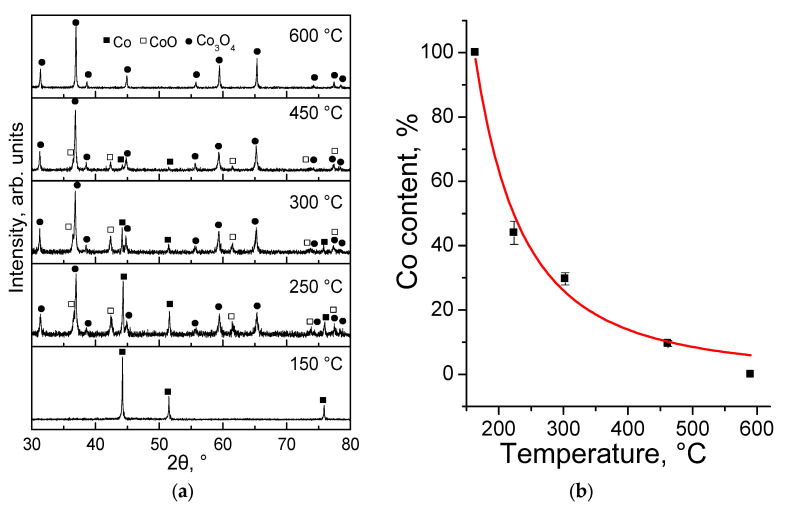
X-ray pattern of cobalt nanoparticles after heating at different temperatures in the air atmosphere (**a**) and change in the Co content in the samples depending on the calcination temperature (**b**).

**Figure 4 nanomaterials-12-02523-f004:**
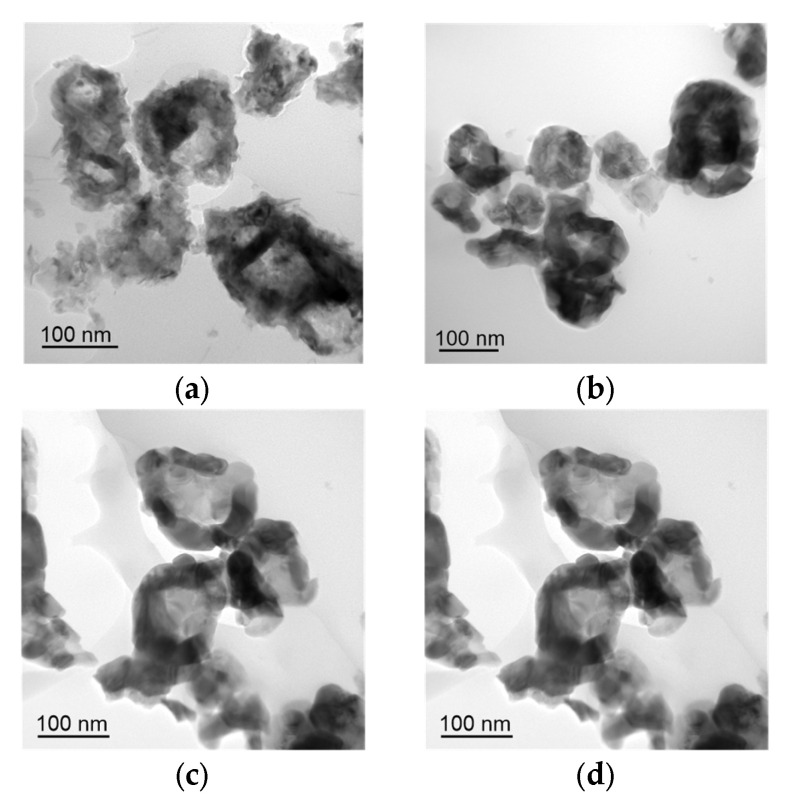
TEM images of the nanoparticles obtained after heating Co nanoparticles up to 250 °C (**a**), 300 °C (**b**), 450 °C (**c**), and 600 °C (**d**).

**Figure 5 nanomaterials-12-02523-f005:**
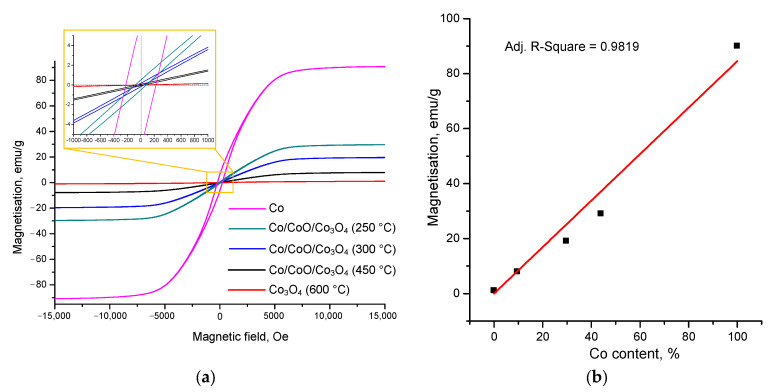
The dependence of saturation magnetisation of Co, Co/CoO/Co_3_O_4_ and Co_3_O_4_ nanoparticles on the magnetic field strength (**a**) and dependence of saturation magnetisation on the Co content in nanoparticles (**b**).

**Table 1 nanomaterials-12-02523-t001:** Quantitative assessment of the content of phases in samples according to X-ray phase analysis.

Sample	The Amount of Substance in the Samples, %		
	Co	CoO	Co_3_O_4_
Co	100	0	0
Co/CoO/Co_3_O_4_ (250 °C)	44.0 ± 3.6	14.7 ± 1.1	41.3 ± 2.9
Co/CoO/Co_3_O_4_ (300 °C)	27.9 ± 1.9	12.1 ± 0.8	60.0 ± 4.2
Co/CoO/Co_3_O_4_ (450 °C)	9.6 ± 0.9	10.2 ± 0.6	80.2 ± 4.4
Co_3_O_4_	0	0	100

**Table 2 nanomaterials-12-02523-t002:** Magnetic characteristics of the samples obtained.

Sample	H_c_, Oe	M_s_, emu/g	M_r_, emu/g
Co	225 ± 11.1	90 ± 3.6	64 ± 3.4
Co/CoO/Co_3_O_4_ (250 °C)	81 ± 5.9	29 ± 2.3	5 ± 3.4
Co/CoO/Co_3_O_4_ (300 °C)	45 ± 3.2	19 ± 2.1	1.7 ± 3.4
Co/CoO/Co_3_O_4_ (450 °C)	54 ± 2.9	7.9 ± 0.7	0.8 ± 3.4
Co_3_O_4_	51 ± 2.7	1.1 ± 0.1	0.1 ± 3.4

**Table 3 nanomaterials-12-02523-t003:** Magnetic characteristics of composites based on cobalt compounds.

Composition of Particles	H_c_, Oe	M_s_, emu/g	M_r_, emu/g	Reference
Co/CoO	600.5	115.5	-	[15]
CoO/Co_3_O_4_	0.2734	3.450	85.032	[20]
Co/Co_3_O_4_	373	127.8	-	[21]
Co/CoO	-	18	-	[22]
Co/Co_3_O_4_	-	59.8	-	[39]
Co/CoO	-	34.6	-	[39]
Co/Co_3_O_4_	46.9	67.7	-	[40]
Co/CoO	-	100.6–74.6	-	[41]
Co/CoO/Co_3_O_4_	198.3–327.3	67.0–24.7	-	[42]
CoO/Co_3_O_4_	330.8	5.8	-	[42]
Co/CoO/Co_3_O_4_	81–54	29–7.9	5–0.8	In this work

## Data Availability

All data generated or analyzed during this study are included in this published article.

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
