# Peer review of "Controlled Oxidation of Cobalt Nanoparticles to Obtain Co/CoO/Co3O4 Composites with Different Co Content"

_nanomaterials, 2022, doi:10.3390/nano12152523_

Round 1

Reviewer 1 Report

The present manuscript describes the investigation of cobalt nanoparticles and their aerobic oxidation process. The work is well written, the methodologies are correct. Oxidation process is monitored by DSC-TG and X-ray phase analysis, which are common methods in this case. Morphology of nanoparticles at different oxidation levels is studied by TEM microscopy. Remarkably, magnetic properties of nanoparticles were investigated, the respective results agree with other methods. In my opinion, the work can be recommended for publication after minor correction: please mention the argon pressure and purity, as well as cobalt purity, in the protocol for Co nanoparticles preparation (“Materials and Methods” section).

Author Response

We appreciate the reviewer for the constructive criticism. We also thank the reviewer for the effort and time put into the review of the manuscript. Major points are highlighted in the text.

Comments and Suggestions for Authors

The present manuscript describes the investigation of cobalt nanoparticles and their aerobic oxidation process. The work is well written, the methodologies are correct. Oxidation process is monitored by DSC-TG and X-ray phase analysis, which are common methods in this case. Morphology of nanoparticles at different oxidation levels is studied by TEM microscopy. Remarkably, magnetic properties of nanoparticles were investigated, the respective results agree with other methods. In my opinion, the work can be recommended for publication after minor correction: please mention the argon pressure and purity, as well as cobalt purity, in the protocol for Co nanoparticles preparation (“Materials and Methods” section).

Response: We have added Argon and Cobalt wire specifications to the “Materials and Methods” section.

Reviewer 2 Report

The authors studied the characteristics of composite Co/CoO/Co3O4 nano-particles formed as a result of oxidation of Co nanoparticles, particularly analyzing the influence of their composition on the magnetic characteristics of the composite. Even though contents and characterization are not particularly novel, the obtained results are based on original hypothesis and clearly presented as well. Therefore, I recommend the submitted manuscript for its publication after some minor revisions:

Please expand the description of the paper purpose within the introduction, for example explaining why your characterization could be useful to tune future Co-based nanocomposites and in which kind of application;

In discussing Figure 3, please indicate the ICDD Cards related to the identified crystalline phases (i.e. Co, CoO and Co3O4);

Try adding a new Table containing an estimation (in %) of the different phases in samples calcined at different temperatures (as you did just for Co in Figure 3b);

Add a description when labelling both Table 1 and Table 2;

In Table 2, you should add also your results (the ones already presented in Table 1) for an easier comparison with literature data.

Author Response

We thank the reviewer for the effort and time put into the review of the manuscript. Each comment has been carefully considered point by point and responded.

Comments and Suggestions for Authors

The authors studied the characteristics of composite Co/CoO/Co3O4 nano-particles formed as a result of oxidation of Co nanoparticles, particularly analyzing the influence of their composition on the magnetic characteristics of the composite. Even though contents and characterization are not particularly novel, the obtained results are based on original hypothesis and clearly presented as well. Therefore, I recommend the submitted manuscript for its publication after some minor revisions:

Please expand the description of the paper purpose within the introduction, for example explaining why your characterization could be useful to tune future Co-based nanocomposites and in which kind of application;

Response: Description of the purpose of the paper added to the introduction

In discussing Figure 3, please indicate the ICDD Cards related to the identified crystalline phases (i.e. Co, CoO and Co3O4);

Response: The ICDD Cards added.

Try adding a new Table containing an estimation (in %) of the different phases in samples calcined at different temperatures (as you did just for Co in Figure 3b);

Response: New table added.

Add a description when labelling both Table 1 and Table 2;

Response: Description of Table 1 and Table 2 added.

In Table 2, you should add also your results (the ones already presented in Table 1) for an easier comparison with literature data.

Response: The results obtained in the work added to table 3.

Reviewer 3 Report

In this manuscript, the authors focused on the study of the patterns of oxidation with air oxygen by heating Co nanoparticles obtained by electrical explosion of wire. There is no novelty in your work and the developed materials should be used for a specific application. Hence, I recommended a major review and it is accepted for this journal after the author clarifies the following comments.

1- Introduction; many statements lack appropriate references. In general, referencing has not been taken into account in many parts such as introduction and discussions.

2- The similar studies should be cited and discussed to highlight the importance and novelty of the developed system.

3- The authors may need to briefly address the difference(s) between the current manuscript and other similar published articles in the Introduction section. 

4- In general, the references in the introduction are poorly chosen, when compared to the sentences they serve as confirmation for.

5- The authors have developed a new material but they should present a possibility for an application of the materials developed, otherwise there is no novelty in this work.

 6-     Conclusions need to be improved by specifying the discussed important points within this work. In the conclusions, the authors should also provide an outlook of the challenges and potential future directions.

Author Response

We thank the reviewer for the effort and time put into the review of the manuscript. Each comment has been carefully considered point by point and responded. The changes in the text are highlighted.

Comments and Suggestions for Authors

In this manuscript, the authors focused on the study of the patterns of oxidation with air oxygen by heating Co nanoparticles obtained by electrical explosion of wire. There is no novelty in your work and the developed materials should be used for a specific application. Hence, I recommended a major review and it is accepted for this journal after the author clarifies the following comments.

1- Introduction; many statements lack appropriate references. In general, referencing has not been taken into account in many parts such as introduction and discussions.

Response: The novelty of the study lies in the study of the possibilities of obtaining composites based on cobalt and cobalt oxides with different ratios of components using Co nanoparticles obtained by electric explosion of a wire. In the introduction, the possible areas of application of such composites are considered in sufficient detail. We have tried to add additional references to the statements used in the introduction.

2- The similar studies should be cited and discussed to highlight the importance and novelty of the developed system.

Response: The introduction has been revised in accordance with the remark.

3- The authors may need to briefly address the difference(s) between the current manuscript and other similar published articles in the Introduction section. 

Response: The introduction has been revised in accordance with the remark.

4- In general, the references in the introduction are poorly chosen, when compared to the sentences they serve as confirmation for.

Response: The references in the introduction have been improved.

5- The authors have developed a new material but they should present a possibility for an application of the materials developed, otherwise there is no novelty in this work.

Response: Possible applications of the obtained materials are added to the introduction.

 6-     Conclusions need to be improved by specifying the discussed important points within this work. In the conclusions, the authors should also provide an outlook of the challenges and potential future directions.

Response: Conclusions have been supplemented in accordance with the comments.